# An Optimization Routing Algorithm Based on Segment Routing in Software-Defined Networks

**DOI:** 10.3390/s19010049

**Published:** 2018-12-22

**Authors:** Xiaolan Hou, Muqing Wu, Min Zhao

**Affiliations:** School of Information and Communication Engineering, Beijing University of Posts and Telecommunications, Beijing 100876, China; wumuqing@bupt.edu.cn (M.W.); zhaomin@bupt.edu.cn (M.Z.)

**Keywords:** software-defined network, segment routing, routing algorithm, performance evaluation

## Abstract

Software-defined networks (SDNs) are improving the controllability and flexibility of networks as an innovative network architecture paradigm. Segment routing (SR) exploits an end-to-end logical path and is composed of a sequence of segments as an effective routing strategy. Each segment is represented by a middle point. The combination of SR and SDN can meet the differentiated business needs of users and can quickly deploy applications. In this paper, we propose two routing algorithms based on SR in SDN. The algorithms aim to save the cost of the path, alleviate the congestion of networks, and formulate the selection strategy by comprehensively evaluating the value of paths. The simulation results show that compared with existing algorithms, the two proposed algorithms can effectively reduce the consumption of paths and better balance the load of the network. Furthermore, the proposed algorithms take into account the preferences of users, actualize differentiated business networks, and achieve a larger comprehensive evaluation value of the path compared with other algorithms.

## 1. Introduction

Software-Defined networks (SDNs) [1] are becoming an innovative paradigm for next-generation networks. The defining feature of SDNs compared to traditional networks is that they decouple the network control plane from the data plane, simplify network management, and make it more flexible. SDN abstracts the network to shield the underlying complexity and the SDN controller provides simple and efficient configuration for the upper application layer [2]. The SDN controller determines the forwarding path for each flow in the network. OpenFlow [3] is a standardized interface that can communicate with the data plane by the controller. OpenFlow separates control functions from the network devices and maintains the flow table structure on the network devices.

Segment routing (SR) [4] can help optimize the network capabilities of IP and MPLS and enable the network to achieve better scalability. The key idea of SR is to perform routing by a few middle points based on a sequence of logical segments formed between the ingress and egress nodes. And the intermediate switch only needs to know how to reach the middle points and to forward packets. SR [5] avoids the requirement for millions of tag encodings to be stored along each path in each network device, so SR reduces the number of forwarding rules in TCAM [6]. In addition, SR eliminates the complexity of maintaining a mass of forwarding rules, so it is able to reduce overhead and the cost of traffic engineering (TE) [7,8].

In network services, users or networks have different requirements. For example, some users have requirements of providing independent paths to achieve routing separation in their networks. Some users need to avoid special paths in routing, and some price-insensitive users require the minimum delay or higher bandwidth. The combination of SR and SDN can well meet the differentiated network services and quickly set up and implement deployment. Traffic engineering is necessary to improve the efficiency of the network and application performance for operators. The SDN controller does not need to add extra forwarding rules to each individual switch along a path [9]. The SDN controller only needs to code end-to-end routing information into an ordered list of tags and store it in the packet header. The intermediate nodes do not need to run complex signaling protocols to maintain the state information of each flow. So, SR greatly simplifies the complexity of network equipment. SR adds a packet header containing an ordered list of labels in the ingress node of the network edge. The intermediate nodes only need to forward the IP packets according to the ordered lists in the segment header [4]. All the state information of routing is maintained by the ingress node.

With the gradual expansion of the network scale, the complexity of networks is constantly increasing. Therefore, routing algorithms face severe challenges and ordeals. The most common performance indicators of routing calculation are cost, hop count, throughput, load balance, delay, jitter, bandwidth, packet loss rate and so on. The evaluation indicators of traditional routing algorithms are mainly divided into two parts. One is for the requirement of overall service quality of the network, such as hop count, throughput, cost, load balance, bandwidth and so on and the other is for the requirement of quality of data transmission based on business, such as delay, jitter, packet loss rate and so on. Most routing algorithms optimize one or more indicators for the overall service quality of the network or optimize the overall service quality under the condition of satisfying certain transmission quality. Since each business has different requirements on transmission quality, most of the existing routing algorithms are not informed by much research on how different businesses can obtain better transmission with a certain network performance guarantees.

In this paper, we address traffic engineering with SR in SDN. Our proposed algorithms consider the preferences of different users for transmission quality under the condition of satisfying the overall service quality of the network. First, we use the multiple objective particle swarm optimization (MOPSO) algorithm [10] to optimize link weight for path cost and load balance. Then, the first *k* shortest paths are selected from the optimized weighted matrix between the source node and the destination node. Finally, according to the preferences of users, the model takes application service as a guide [11], and the path with the largest comprehensive evaluation value is selected from the first *k* shortest paths. In order to obtain higher accuracy, we also propose an improved MOPSO algorithm for the optimization of link weight in the first part of the algorithm. Our proposed algorithms have better network performance than other traditional routing algorithms such as the shortest path first (SPF) [12], shortest widest path (SWP) [13], widest shortest path (WSP) [14], minimum interference routing algorithm (MIRA) [15], and the algorithm in [16], which we refer to as the Lee algorithm in our paper. Simulations show that our algorithms not only can reduce path consumption, better balance the network load, and decrease the maximum link utilization rate but also obtain a larger comprehensive evaluation value of the path compared with the above algorithms. 

The structure of the paper is as follows: in Section 2, we discuss related work. In Section 3, we introduce our proposed algorithms. In Section 4, the performance evaluation is demonstrated. In Section 5, we complete this paper with a conclusion.

## 2. Related work

### 2.1. Segment Routing Based on SDN

SR [4] is a recently proposed routing architecture to solve the complex problem of a large number of forwarding rules in the information forwarding process. A segment [17] represents an instruction that can be used to define the path in a weighted graph. A node segment contains the unique label of the next switch to reach. The intermediate switch only needs to know how to forward to the next intermediate node with the shortest path. They no longer need to maintain the status of a large amount of network information. In the current draft defined by the IETF, two segments are defined: the Nodal Segment and Adjacency Segment. The Nodal Segment is a global label. Each node is assigned by a globally unique Nodal Segment. The Adjacency Segment is a local label that is locally valid and used to represent a particular segment routing node. 

In Figure 1, we use a simple network topology to clarify the overview of SR based SDN. The SDN controller calculates the explicit route through the routing module and configures the forwarding table of the ingress switch with a segment list. The node only processes the top label of the stack to forward the packets along the shortest path to reach the specified destination. For example, the target path for an incoming traffic flow is from A to F, and the path is an encoded SID list E, 101, F in the packet header. E and F are the Nodal Segment. 101 is the Adjacency Segment. The switch A processes the top label of the stack and forwards the packet to the switch E along the shortest path. Then, switch E pops the top label to forward the packet to F along the shortest path from output port 101. After arriving at switch C, the packet is forwarded to F along the shortest path. SR maintains per-flow path information only at the ingress node, where the label stack is initialized by the SDN controller. In this way, the operation of the control plane and the provisioning procedure of time consumption is remarkably simplified.

Some works have been done on this issue. The authors in [4] present the concept of SR and its implementation. SR uses the source routing method. The idea of SR is to divide an end-to-end path into multiple segments, each of which is identified by a segment identifier. The forwarding steps are designated in the segment, so the routers do not have to maintain routing table information. The header of the packets carries the segment. The node forwards the packet to the destination via the shortest path. The authors note that SR has a unique advantage in implementing network load balance and traffic engineering. As we know, SDN has a global view of the network and has the ability to control the segments. The combination of SR and SDN is a good way to meet these differentiated business needs and can quickly realize the deployment and implementation of business needs. However, our work mainly considers the routing algorithm about SR in SDN.

The authors in [18] aim at solving the problem of restoration optimization in SR. The main problem is to centralize connection to the primary path in order to achieve optimal sharing of recovery bandwidth in non-synchronous network failures. The authors propose linear programming problem and random rounding scheme in this paper. The primal-dual algorithm solves the linear programming problem, and it has a good running time even on large networks. The simple random rounding scheme aims at other restrictions on segment routing. However, our research aims at evaluation and selection of paths for SR in SDN.

The authors in [19] concentrate on two related situations: dynamic traffic recovery and multi-domain traffic engineering. This paper defines the appropriate procedure to minimize the depth of the segment list. With regard to traffic recovery, an SR-FAILOVER scheme is proposed to dynamically recover traffic flow interrupted by link or node failures to minimize the depth of the required segment list. And most backup paths can be encoded with a segment list of one or two labels. In terms of multi-domain traffic engineering, the pd-SR scheme and e2e-SR scheme are proposed, which utilize a two-way communication session established among the SR controllers of the traversed domains. However, the two schemes use different procedures to determine the segment list to be applied to the packets and compare based on the depth of the segment list. However, the authors do not consider the routing selection process for the SDN controller to deploy routing information.

The authors in [20] introduce an integer linear programming (ILP) model to heuristically evaluate the TE performance of SR-based packet networks. Three different ILP models are proposed, the ECMP model, SHP model, and SEGMR model. Because the computational complexity of the ILP model is high, some instances require too much time to be resolved. Thus, the authors propose a heuristic approach to determine the unique route between each pair of nodes that need to transmit information. Heuristic attempts to keep the lowest possible network utilization at a minimum and to ensure that the maximum value of the SLD is not exceeded. Our work, however, focuses on a routing algorithm that considers the optimization of link weight and the evaluation of paths.

The authors in [16] propose a routing algorithm for SDN with SR that can satisfy the bandwidth demand of routing requests. This algorithm takes into account the load balance and the additional cost of packet header size reduction in the network. The algorithm uses the criticality and congestion index of links to define the link weight. Then, the authors design the Bellman-Ford algorithm with a hop count restriction to solve the problem of minimum weight path. The proposed algorithm takes into account the additional network overhead and network resource through the constraint of path length. However, the authors do not consider user preferences and the actual network situation. Our work not only designs the optimization of link weight for the path consumption and load balance of the network but also takes the network application service as a guide, realizes the reasonable evaluation of path performance, and finally selects the best path.

### 2.2. Traffic Engineering for Segment Routing in SDN

Traffic engineering refers to a series of methods to optimize the performance of a network [8]. The goal of traffic engineering is to dynamically analyze and predict the data flow in the network. For different types of networks, traffic engineering technology has been evolving to meet different needs. With the development of network technology, users have higher performance and reliability requirements for the application services of networks. So, the stability and cost of networks are very important. SDN shows its advantage in effectively solving this problem. SDN provides a better foundation for traffic engineering. First, SDN provides a centralized global view, and the controller can get the global network status in real time. Second, SDN provides a programmable data layer interface, and the operator can dynamically redistribute the network resource. In addition, the switches of multiple device vendors use a unified programming interface to provide full openness. The idea of SR is to divide an end-to-end path into multiple segments. Each segment is identified by an SID [21]. If the network state changes or the policy needs to be changed, the SDN controller recalculates the path and updates its segmented list at the previous first hop or segment changed router. 

Our work first focuses on optimizing the link weight for saving path cost and load balance. Then, it chooses the first *k* shortest paths to get a shorter segment list. Because a longer path means a longer segment list, more energy consumption. Finally, we use the comprehensive evaluation model to select the path with the best comprehensive evaluation value from the first *k* shortest paths, that is, the best path for comprehensive performance. First, we introduce several traditional routing algorithms. The shortest path first (SPF) [12] algorithm uses the shortest path algorithm to calculate the path. The SPF algorithm does not consider if the bandwidth of the selected path satisfies the need of the actual application. The idea of the widest shortest path (WSP) [14] is to find a path with the largest remaining bandwidth in all min-hop paths. The shortest widest path (SWP) algorithm [13] is to find the shortest path with the largest remaining bandwidth. The idea of the minimum interference routing algorithm (MIRA) [15] is to minimize the impact on the establishment of link requests for future nodes. Lee and Sheu in [16] propose a routing algorithm for SDN with SR. The algorithm can meet the bandwidth requirement of routing requests and reduce the extra cost of packet header size. However, the algorithm does not consider preferences of users to evaluate the link performance of a network.

## 3. Algorithm Design

### 3.1. Proposed Optimization Algorithm for Link Weight

With the increase in traffic in the network, if the traffic cannot be reasonably allocated, it is possible that some links may be congested, whereas other links are not fully utilized. In order to make full use of network resources, the setting of link weight is decisive for path cost and load balance. In this paper, we use multi-objective particle swarm optimization (MOPSO) algorithm [10] to optimize the link weight for path cost and load balance. We set up multiple populations to find the optimal solution through the evolution of collaboration between populations. The two objective functions are used to measure path cost and load balance, respectively. Then, we calculate the first *k* shortest paths between the source node and the destination node. Finally, an optimal path is selected from the first *k* shortest paths according to the comprehensive evaluation model. We refer to the whole procedure above as Algorithm 1. In addition, we propose an improved MOPSO algorithm in order to better optimize the link weight in the first step. And we call Algorithm 2.


**Algorithm 1: Find the optimal path *p***
1. Input: objective function fitness1, fitness2, source node and destination node2. Traffic matrix ***G*** is updated by SDN controller after a certain number of requests3. Weighted matrix ***G****1* ← MOPSO (fitness1, fitness2, ***G***, source, destination)4. Find the first *k* shortest paths from the source to the destination based on the weight matrix ***G****1*5. Find the optimal path *p* according to the comprehensive performance evaluation model.6. Output: the path *p*


**Algorithm 2: Find the optimal path *p***
1. Input: objective function fitness1, fitness2, source node and destination node2. Traffic matrix ***G*** is updated by SDN controller after a certain number of requests3. Weighted matrix ***G****1* ← the improved MOPSO (fitness1, fitness2, ***G***, source, destination)4. Find the first *k* shortest paths from the source to the destination based on the weight matrix ***G****1*5. Find the optimal path *p* according to the comprehensive performance evaluation model.6. Output: the path *p*

The particle swarm optimization (PSO) algorithm is a group of intelligent algorithms that achieves optimization through collective collaboration between birds. The MOPSO algorithm is used to solve the multi-objective optimization problem [22] and set up multiple populations. The optimal solution is found by the cooperative evolution between populations. The main steps of the MOPSO algorithm are as follows:
(1)Initialize the population of particles, and customize the particle initial velocity and position;(2)According to the objective function, determine the value of objective function of each particle initialized;(3)According to the definition of the Pareto dominance relationship, the non-inferior solution is stored in an external archive;(4)Select the global optimal value from the external archive;(5)Update the velocity and position of the particle at the next moment by the updating formula of velocity and position;(6)Calculate the new fitness of particles;(7)Update the individual extremum and global extremum;(8)Update the external archive;(9)If the termination conditions are not satisfied, then jump to the third step and then run.

Recently, various improved approaches to MOPSO have been developed, e.g., in [23,24,25,26]. Wu et al. [23] propose a MOPSO algorithm with asynchronous updates. When a particle in the swarm is regulated, the Pareto front is immediately evaluated. New positions of the subsequent particles not only depend on the message of the previous iteration but are also partially based on the optimal solutions of the current iteration. In [24], aiming at multi-objective large-scale optimization, many-objective optimization, and distributed parallelism, the authors propose some methodologies and discuss other prospective research trends. In [25], multi-objective optimization problems (MOPs) are transformed into a set of sub-problems based on a decomposition approach, and then each particle is allocated according to each optimized sub-problem. Han et al. [26] propose an adaptive MOPSO (AMOPSO) algorithm based on a hybrid framework to solve the distribution entropy and population spacing (SP) information. The proposed algorithm improves the search performance in terms of convergent speed and precision.

In this paper, three improvements are put forward based on the AMOPSO algorithm:

(1) The solution distribution entropy at the *t*th iteration is calculated by the probability distribution function of the cell with non-dominated solutions in the archive. The variation in distribution entropy determines the evolutionary tendency of the population. However, the method is only suitable for the optimization of two goals. We use a parallel cell coordinate system (PCCS) [27] to calculate the distribution entropy, which can be used for multi-dimensional data analysis and visualization. The difference in entropy represents the change in the approximate Pareto front redistribution, which is then used to assess the diversity and convergence and thus infer the potentiality of finding a new solution to the current population. The Pareto entropy and the difference in entropy change greatly, indicating that the new solution dominates a large number of old solutions in the external archive. So, this leads to a wide range of approximate Pareto front changes, and the convergence of the approximate Pareto front needs to be enhanced. The smaller the difference in entropy, the smaller the range of the approximate Pareto front redistribution. The diversity of the Pareto front needs to be enhanced.

(2) In [26], the convergence degree is evaluated by the dominate strength of the *i*th solution at the *t*th iteration and the difference between solution and dominated solutions. In our algorithm, we use a more intuitive way to evaluate the convergence of solutions. The essence of convergence performance is to calculate the average distance of the target vector from the optimal Pareto front to the real Pareto; thus, the average distance obtained is used to measure the degree of proximity of the final solution and the non-dominated optimal solution. So, we evaluate the convergence of solutions by finding the distance between each solution and the optimal Pareto frontier.

(3) The authors in [26] do not evaluate the diversity of solutions in detail. In this paper, we use the distribution of indicators, namely, the variance of the distance between adjacent target vectors, to measure the diversity of non-dominated solutions obtained. The smaller the variance is, the better the diversity.

$f_{k,m}$ denotes the *m*-th target value of the *k*-th non-dominant solution in the Pareto optimal solution set, and $f_{k,m}$ can be mapped into a two-dimensional planar grid with *K* × *M* cells according to (1):
(1)$$L_{k,m}=\left\lceil K\frac{f_{k,m}-f_{m}^{\min}}{f_{m}^{\max}-f_{m}^{\min}}\right\rceil$$
where ⌈x⌉ is the rounding function; *k* = 1, 2, ..., *K*, *K* is the number of members in the current iteration for external archive. *m* = 1, 2, ..., *M*, *M* is the problem of the number of goals to be optimized; $L_{k,m}\in \{1,2,\ldots ,K\}$ is an integer number in which $f_{k,m}$ is mapped to PCCS. The data set on the Pareto front can all be mapped into the PCCS by using (1), as shown in Figure 1. $P_{k,m}$ represents the *m*-th cell coordinate component of the *k*th target vector. Figure 2 shows the parallel cell coordinate system of five cell coordinates and three objective functions.

During the *t*-th iteration, the Pareto entropy *Ent*(*t*) of the approximate Pareto front stored in the external archive can be calculated according to (2):
(2)$$Ent(t)=-\sum_{k=1}^{K}\sum_{m=1}^{M}\frac{C_{k,m}(t)}{KM}\log\frac{C_{k,m}(t)}{KM}$$
where $C_{k,m}(t)$ denotes the number of cell coordinate components falling to the *m* column of the *k*-th row after the approximate Pareto front is mapped to the PCCS.

In the two adjacent iterations *t* − 1 and *t*, Δ*E*(*t*) denotes the difference in entropy, which indicates the degree of change in the approximate Pareto front. Δ*E*(*t*) can be calculated by (3):
(3)ΔE(t)=E(t)−E(t−1)

The global optimal solution is given as:
(4)$$\mathrm{gbest}(t)=\left\{ \begin{array}{l} \mathrm{dgbest}(t)\;|\Delta E(t)|<\delta_{c}\\ \mathrm{cgbest}(t)\;|\Delta E(t)|>\delta_{c} \end{array}\right.$$
where dgbest(*t*) is the solution with the best diversity at the *t*-th iteration; cgbest(*t*) is the solution with the best convergence at the *t*-th iteration.(5)$$\mathrm{dgbest}(t)=\arg\min DV(x_{i}(t))$$
where $DV(x_{i}(t))$ denotes the diversity degree of the *i*th solution $x_{i}(t)$.
(6)$$DV(x_{i}(t))=|\bar{d}-d_{i}|$$
where $\bar{d}$ is the average value of all $d_{i}$, and the operation formula of $d_{i}$ is as follows:
(7)$$d_{i}=\underset{j=1,\ldots ,|P|,j\neq i}{\min}\left\{\sum_{m=1}^{M}\left|f_{m}(x_{i})-f_{m}(x_{j})\right|\right\}$$
where *M* represents the number of objective functions; *P* represents the best solution set on the approximate Pareto front; $f_{m}(x_{i})$ represents *m*-th objective function value of $x_{i}$.
(8)$$\mathrm{cgbest}(t)=\arg\min CV(x_{i}(t))$$
where $CV(x_{i}(t))$ is the convergence degree of the *i*-th solution. $CV(x_{i}(t))$ is defined as:
(9)$$CV(x_{i}(t))=\underset{1\leq j\leq |P^{*}|}{\min}\sqrt{\sum_{m=1}^{M}(\frac{f_{m}(x_{i})-f_{m}(x_{j}^{*})}{f_{m}^{\max}-f_{m}^{\min}}{)}^{2}}$$
where $P^{*}=\{x_{1}^{*},x_{2}^{*},\ldots ,x_{\left|P^{*}\right|}^{*}\}$ represents the best solution set on the real Pareto front.

In the following part, we set the objective functions. We use a weighted graph G=(V,E) to present the network topology of SDN, where ***V*** represents the set of nodes and ***E*** represents the set of edges. For each link, there is a link capacity *c*(*e*) and weight *w*(*e*). Let *f*(*e*) denote the total amount of traffic flows carried by the link *e*. Thus, u(e)=f(e)/c(e) denotes the link utilization of the link *e*. The goal of traffic engineering is to maintain u(e)≤1 for any e∈E, but this goal is too general. We define the function in (10), which describes the link cost:
(10)$$\psi (e)=\left\{ \begin{array}{ll} u(e), & 0\leq u(e)<\frac{1}{3}\\ 3u(e)-\frac{2}{3}, & \frac{1}{3}\leq u(e)<\frac{2}{3}\\ 10u(e)-\frac{16}{3}, & \frac{1}{3}\leq u(e)<\frac{9}{10}\\ 70u(e)-\frac{178}{3}, & \frac{9}{10}\leq u(e)<1\\ 500u(e)-\frac{1468}{3}, & 1\leq u(e)<\frac{11}{10}\\ 5000u(e)-\frac{16318}{3}, & \frac{11}{10}\leq u(e)<\infty \end{array}\right.$$

According to (10), we can see that the cost increases faster and faster with the growth in link utilization. When the link utilization rate is close to 1 (the link is congested), the cost rises sharply. A link with high link utilization rate has a high cost, which imposes a punitive high cost on the overloaded link. Therefore, when seeking a path, the allocation scheme to avoid link overload will be chosen. The cost of a path is not only related to link utilization but also to the length of the path. We take the sum of the cost of all the links on a path as the cost of this path. The cost function of path *p* is defined as follows [28]:
(11)$$\cos t(p)=\sum_{e\in p}\psi (e)$$

The variance of the link utilization expresses the load balance. The smaller the variance, the more balanced the load distribution is. Let $\bar{u}$ denote the average link utilization. The formula of load balance is expressed as:
(12)$$\sigma =\sum_{e\in p}{\left(u(e)-\bar{u}\right)}^{2}/h(p)$$
where *h*(*p*) denotes the hop counts of path *p*. So, we set two objective functions as follows:
(13)$$fitness1=\sum_{e\in p}\psi (e)$$
(14)$$fitness2=\sum_{e\in p}{\left(u(e)-\bar{u}\right)}^{2}/h(p)$$

As SDN has a global view of the network resource, it is able to get the global network traffic to formulate a better strategy for network optimization. Under the condition of a given traffic demand matrix from the SDN controller, link weight determines the shortest path between the source node and the destination node and further influences the path cost and load distribution in the network. Therefore, the optimization of link weight has become an effective way to realize traffic engineering. In this paper, we use the path cost and load balance of the network as the two optimization objectives for the MOPSO algorithm and improved MOPSO algorithm. The improved MOPSO algorithm for optimization of link weight is summarized in Algorithm 3. The traffic matrix G1 can be updated by the controller after each routing request.


**Algorithm 3: Improved MOPSO algorithm**

Input: objective function fitness1, fitness2, traffic matrix ***G***, source node and destination node.Initializing the population size, the positions and velocity of particles, the flight parameters, the maximal size of archive, and maximal iterations.While iteration <= maximal iterationsCalculate the value of fitness1 and fitness2.Calculate the non-dominated solutions and update the archive.If the number of archive > maximal sizePruning the archiveEndThe external archive is mapped to PCCS.Calculate the entropy and the difference of entropy.Choose the *gbest* according to the difference of entropy.Update the flight parameters. Update the positions and velocity of particles.EndOutput: the best position, namely weighted matrix ***G****1*


Through the MOPSO algorithm or improved MOPSO algorithm, the path cost and load balance are optimized. After calculating the weight of the traffic matrix, the first *k* shortest paths from the source and the destination node are calculated. This reduces the length of the sub-list to decrease the consumption of the packet header to a certain extent. Then, the optimal path is selected from the first *k* shortest paths according to the comprehensive evaluation model.

### 3.2. Comprehensive Evaluation Model

The optimization of link weight by the MOPSO algorithm or improved MOPSO algorithm ensures network service quality in terms of path cost and load balance. Next, we fully consider the different requirement of different businesses for transmission quality and evaluate path quality through the hierarchical comprehensive evaluation model. The comprehensive evaluation model has been applied in many fields [29,30], and the performance is evaluated comprehensively by establishing a hierarchical model according to multiple indicators [31,32]. In order to provide users with highly reliable network service, it is useful to establish an effective and feasible evaluation method for the network application service. In the evaluation process of network performance, the network evaluation is mainly based on many indexes such as delay, jitter, and packet loss rate. However, different indicators cannot be compared uniformly. So, this cannot fully reflect the operation of the measured network. In this paper, we use a comprehensive evaluation model for network service performance [33]. The model takes the preferences of users into account and takes the application service of the network as the orientation. First, we set up an evaluation hierarchy and establish the criterion weight and the scheme weight in the hierarchical structure. Then, the actual measurement data are normalized. Finally, the fuzzy analytic hierarchy process (AHP) is used to evaluate the performance of each service in the target path.

#### 3.2.1. Establish the Hierarchical Structure of Evaluation of Network Application Business 

In this paper, we establish a three-layer evaluation model for network applications. We select the evaluation key performance index (KPI) set for delay, jitter, and packet loss rate for the QoS evaluation criteria. Thus, it overcome the one-sidedness of a single evaluation index and achieve multi-index fusion. The layer of the business scheme is divided into three network applications G = {web interactive service, voice communications service, streaming media service}. Figure 3 shows the hierarchical structure of the network performance evaluation model based on application service.

#### 3.2.2. Calculate the Scheme Weight and the Criterion Weight

In the fuzzy AHP hierarchy, the importance of different evaluation indicators of the same business is relevant, so we adopt the method of pairwise comparison to judge the matrix. But the values of different business programs are independent of each other. Therefore, we divide the scheme weight and the criterion weight separately in the fuzzy AHP. The weight of the business scheme is calculated by user preference or actual demand, but the criterion weight is still obtained through the establishment of the fuzzy judgment matrix.

First, we determine the scheme weight. According to user preference or the network application, three weight values of the network application are given. Different users or business needs lead to different weight values of the network business. 

Then, we determine the criterion weight. Different network applications have different levels of sensitivity to each index. The business of voice communication requires data latency and jitter to be as small as possible; the real-time streaming media business has certain requirements for bandwidth and packet loss rates; the web interactive business requires a rapid response network. To provide differentiated service and high network availability, the evaluation performance of a network is necessary for each network application business. The specific steps are as follows:

(1) Establish a fuzzy judgment matrix $R={\left(r_{ij}\right)}_{n\times n}$, where $r_{ij}$ represents the relevance between the *i* element and the *j* element in this hierarchy. We use the number in the range of 0.1 to 0.9 to quantitatively describe the relative importance of any two QoS indicators. The greater the value of $r_{ij}$
$\left(0.1\leq r_{ij}\leq 0.9\right)$, the greater is the importance of the *i* element than the *j* element.

(2) Transform into a fuzzy consistency matrix. First, according to (15), we calculate the sum of the rows of the judgment matrix, that is:
(15)$$\xi_{i}=\sum_{j=1}^{m}r_{ij};\;i=1,2,\ldots ,n$$

Then, we calculate fuzzy consistency matrix ***Q*** by using (16):
(16)$$\xi_{ij}=\frac{\xi_{i}-\xi_{j}}{2n}+0.5$$

(3) Based on the normalization process of the sum of the rows, the weight vector is calculated by using (17):
(17)$$w_{i}=(\sum_{j=1}^{m}q_{ij})/(\sum_{i=1}^{n}\sum_{j=1}^{m}q_{ij});\;i=1,2,\ldots ,m$$

#### 3.2.3. The Indicators Are Normalized

We measure the delay, jitter, and packet loss rate of the first *k* shortest paths after calculating the weighted optimal matrix. In the system of the quality evaluation index of the network, the index value of the different classes is standardized by using (18):
(18)$$d_{ij}=\left\{ \begin{array}{ll} \frac{x_{i}^{\max}-x_{ij}}{x_{i}^{\max}-x_{i}^{\min}}, & x_{i}^{\max}-x_{i}^{\min}\neq 0\\ 1, & x_{i}^{\max}-x_{i}^{\min}=0 \end{array}\right.$$

#### 3.2.4. Calculate the Value of Performance Evaluation

$A={\left(a_{ij}\right)}_{3\times k}$ denotes the value of the performance evaluation matrix of the three types of network business at each moment of each link $a_{ij}$ is given by:
(19)$$a_{ij}=W_{i}\times D_{j}^{T};\;0\leq a_{ij}\leq 1$$
where *W_i_* (*i* = web, voice, stream) represents the weight vector of the *i* business scheme; and $D_{j}^{T}$ represents the vector of function fraction of *j* link at the corresponding time.

#### 3.2.5. Calculate the Performance Evaluation of Entire Network

The value of the performance evaluation of the whole network is calculated according to the preference of weight vector of the three types of application business. The result of comparison between the paths is obtained according to (20):
(20)$$P_{j}=W^{\prime }\times A^{T};\;0\leq P_{j}\leq 1$$
where *P_j_* represents the assessed value of total network performance of path *j*; *W*′ represents the weight vector of the business scheme; *A* represents the vector of the assessed value of the business performance of the path in the web interaction, voice communication, and multimedia service.

### 3.3. Algorithm Summary

In this paper, we focus on the evaluation of path quality in order to get a better transmission quality under the condition of satisfying the overall service quality of the network. So, we combine the optimization of link weight with the comprehensive evaluation of paths to find the best path for SR in SDN. In order to avoid partial link congestion caused by excessive energy consumption and the unbalance of traffic distribution in the network, we take the path cost and load balance as the two optimization objectives to establish the optimization algorithm model and apply the MOPSO algorithm or our proposed improved MOPSO algorithm to solve the optimal link weight. Then, in order to guarantee the specific need of the business performance of the users or the actual requirement, we use a performance assessment model to make a comprehensive assessment from the first *k* shortest paths to find the path of best performance. For example, there is a routing request whose source address is A and destination address is B. The SDN controller first uses the improved MOPSO algorithm to optimize the weight of the link, and make the path cost and load balancing index in the network smaller. Then the first *k* shortest path is calculated. Assuming that user uses Web interactive services to book online tickets, user requires the network to respond quickly, minimizes network delay and packet loss rate. At this time, the weight of the scheme can be set to $W^{\prime }=\left[1,0,0\right]$. Then the fuzzy judgment matrix can be established according to the importance of network parameters to determine the weight vector of the criterion. Through the measurement of the indicators of *k* paths (bandwidth, jitter, packet loss rate), the indicators can be standardized, and the service performance evaluation value of *k* paths can be calculated. That is, an optimal path is chosen according to the user’s requirement for service and the network performance index of *k* paths.

The steps of Algorithm 1 are as follows: First, we use the MOPSO algorithm to optimize the link weight for link cost and load balance. Then, we find the first *k* shortest paths. Finally, we use the comprehensive evaluation model to find the best path. Furthermore, in order to make the optimization result more accurate, we propose an improved MOPSO algorithm in the first step of Algorithm 2. The MOPSO algorithm and the improved MOPSO algorithm both need multiple iterations to update the velocity and position of a large number of particles to find the optimal solution. If the size of the archive is not restricted, all non-inferior solutions that satisfy the basic conditions can be entered into the archive. However, as the archive is updated in each iteration, the disadvantage of establishing the archive according to the MOPSO algorithm is that the size of archive is too large and the cost of calculation is increased. In the worst case, if the location of all the particles found in every generation can be entered into the archive, the computing complexity of each generation is *O*(*mN*^2^), where *m* is the number of objects, and *N* is the size of the population. So the calculation complexity of running an algorithm is *O*(*mT_max_N*^2^), of which *T*_max_ is the largest iteration. The improved MOPSO algorithm restricts the size of the archive. When the size of the archive reaches a predetermined upper limit, it is necessary to delete some non-inferior solutions in the archive and reduce the computational complexity to a certain extent. So the computational complexity of two algorithms is large. And the two algorithms need a longer running time than other traditional algorithms when link weights are optimized. So, our proposed algorithms are more suitable for systems that require more accurate experimental results, better equipment performance, and real-time requirements that are not too strong. How to design a multi-target algorithm with lower complexity and faster speed is also a problem to be solved in the future. As the computing power of hardware devices increases, the problem of large computational complexity will be solved. Moreover, with the expansion of the network scale, the deployment of distributed multi-controllers architecture in SDN will solve the problem of large control plane overhead. Our proposed routing algorithms in the paper is designed for SR in SDN. Of course, our method can also be applied to traditional distributed networks and networks not using SR. In the traditional distributed networks, our proposed routing algorithm can be applied in the route calculation module of router by online or offline. 

## 4. Performance Evaluation

In this section, we compare the performance of our proposed algorithms with the Lee algorithms in [16], SPF [12], WSP [14], SWP [13], MIRA [15] and in terms of link cost, load balance, maximum link utilization and the value of path evaluation. In this paper, we use MATLAB to simulate and randomly generate the Waxman network topology. The probability of having an edge between node *u* and node *v* is given by formula $\beta e^{-\frac{d}{\alpha *len}}$, where α and β are the network characteristic parameters. The larger the α, the greater the proportion of the short side relative to the long side. The larger the β, the greater the density of the edge. *d* is the Euclidean distance between any two nodes in the graph, and *len* is the length of a side of the square area. The generated Waxman topology is 80 nodes with 602 links, 100 nodes with 943 links, and 150 nodes with 2190 links. In the simulation, each source host sends the flow packets to the destination host and each flow packet is in the range of 10 MB to 100 MB. Routing requests are sent 10 times per second [34]. The link capacity is set to 1 GB/s. We take *k* = 5 when calculating the first *k* shortest paths.

### 4.1. Calculate the Value of Comprehensive Evaluation

In the comprehensive evaluation model, we set the fuzzy complementary judgment matrix of the three kinds of network applications as $R_{\mathrm{web}}$, $R_{\mathrm{voice}}$, and $R_{\mathrm{stream}}$, respectively. We define their values as follows:
(21)$$R_{\mathrm{web}}=\left[\begin{array}{l} 0.5,0.2,0.2\\ 0.8,0.5,0.5\\ 0.8,0.5,0.5 \end{array}\right];\;R_{\mathrm{voice}}=\left[\begin{array}{l} 0.5,0.5,0.7\\ 0.5,0.5,0.8\\ 0.3,0.2,0.5 \end{array}\right];\;R_{\mathrm{stream}}=\left[\begin{array}{l} 0.5,0.4,0.4\\ 0.6,0.5,0.3\\ 0.6,0.7,0.5 \end{array}\right];$$

Then, we calculate the corresponding consistency matrix $Q_{\mathrm{web}}$, $Q_{\mathrm{voice}}$, $Q_{\mathrm{stream}}$ and weight vector $W_{\mathrm{web}}$, $W_{\mathrm{voice}}$, $W_{\mathrm{stream}}$ according to (15)–(17):
(22)$$Q_{\mathrm{web}}=\left[\begin{array}{l} 0.5000,0.3500,0.3500\\ 0.6500,0.5000,0.5000\\ 0.6500,0.5000,0.5000 \end{array}\right];\;Q_{\mathrm{voice}}=\left[\begin{array}{l} 0.5000,0.4833,0.6167\\ 0.5167,0.5000,0.6333\\ 0.3833,0.3667,0.5000 \end{array}\right];\;Q_{\mathrm{stream}}=\left[\begin{array}{l} 0.5000,0.4833,0.4167\\ 0.5167,0.5000,0.4333\\ 0.5833,0.5667,0.5000 \end{array}\right];$$
(23)$$W_{\mathrm{web}}=[0.2666,0.3667,0.3667];\;W_{\mathrm{voice}}=[0.3555,0.3667,0.2778];\;W_{\mathrm{stream}}=[0.3111,0.3222,0.3667];$$

In the experiment, we randomly select the source and destination nodes for each packet request, so each different path leads to different delay, jitter and packet loss rate. We randomly select a set of data as an example in Table 1.

These measurements are normalized by using (18). We can derive the result vector of the link performance index based on the network application service as follows:
(24)D1=[1,1,1]; D2=[0.70,0.69,0.70]; D3=[0.80,0.79,0.80]; D4=[0.83,0.84,0.83]; D5=[0.62,0.61,0.61];

The value of the overall performance evaluation of the entire network of each link is calculated by using (20): P=[1,0.70,0.79,0.83,0.61]. So, it is clear that the value of overall performance evaluation of the first path is the highest.

### 4.2. Simulation Results

Figure 4, Figure 5 and Figure 6 show a performance comparison of the algorithms when the size of the network is 100 nodes. Figure 4 shows the path costs for SPF, WSP, SWP, MIRA, the Lee algorithm and our proposed two algorithms. The paths cost in the network represents the sum of the consumption of all paths in the corresponding number of requests. The cost of a path obtained by using (11) is the sum of the cost of each link on a path. So, the paths cost depends not only on link utilization but also on the length of the end-to-end path. As shown in Figure 4, the paths cost of the MIRA algorithm is the largest. Because the MIRA algorithm does not consider congestion on non-critical links, large link utilization causes excessive link cost. And the MIRA algorithm avoids the critical links that tend to cause longer paths. The paths cost of SPF is lower than that of MIRA. The SPF usually selects the shortest path, although the algorithm does not consider the remaining bandwidth of the links. SWP and WSP perform better than SPF because they take into account the hop counts as well as the remaining bandwidth of the links. SWP selects the minimum hops path with the largest available bandwidth. WSP selects the largest available bandwidth with the minimum hops paths. SWP has a longer path than WSP. So, the paths cost of the WSP algorithm is less than that of the SWP algorithm. The Lee algorithm takes into account the link criticality and link congestion index. The number of hops of the path is limited. The Lee algorithm performs better than other traditional algorithms, but it does not consider the consumption of the links resource and load balance in the network. Our proposed Algorithm 1 takes the path cost as an optimization goal to achieve the lowest path cost compared to the above five algorithms. The proposed Algorithm 2 makes some improvements in obtaining the global optimal solution and shows the best performance for paths cost.

Figure 5 shows the load balance of the network with the seven algorithms. We define the variance of the link utilization on the network as the load balance by using (12). As shown in Figure 5, the performance of SPF for load balance is the worst without considering link utilization. The performance of WSP is better than SPF because WSP selects the path with the largest remaining bandwidth when it has multiple shortest paths. WSP gives priority to the number of hops and SWP gives priority to the residual bandwidth of paths. The performance of SWP is better than WSP in terms of load balance. The main idea of MIRA is to minimize the influence of future nodes to build the link request and thus effectively avoid the congestion of the critical links. The performance of the MIRA algorithm is better than that of SPF, SWP, and WSP. The Lee algorithm considers the link congestion and has a good effect on load balance. Our proposed Algorithm 1 takes load balance as an optimization target and has obtained better performance than the other five algorithms through multiple iterations of the MOPSO algorithm. Our proposed Algorithm 2 finds a better solution and has the best effect by improving the selection mechanism.

Figure 6 shows the maximum link utilization of the network for the seven algorithms. The smaller the maximum link utilization, the greater is the remaining bandwidth, and the better the performance is in avoiding congestion. Because the SPF is the shortest path algorithm without considering available bandwidth, it is easy to create link congestion when the traffic in the network gradually increases, so its maximum link utilization is the largest. However, the WSP selects the path with the largest bandwidth among the shortest paths. So the maximum link utilization of WSP is lower than SPF. The selection process of SWP is just the opposite of WSP and takes the remaining bandwidth of the path as the first consideration, so the maximum link utilization of SWP is lower than WSP. MIRA chooses the minimum interference path and reduces the blocking rate of the service, which helps improve the utilization rate of the network resource. As a result, its maximum link utilization rate is lower than SWP. Our proposed Algorithm 1 optimizes the link weight through a multi-objective optimization algorithm and achieves lower link utilization than the above four algorithms. However, the Lee algorithm considers the link congestion index when optimizing link weight and avoids link congestion. So, the maximum link utilization of the Lee algorithm is lower in our proposed Algorithm 1. However, the proposed Algorithm 2 makes some improvements in the selection of global optimal particles so that a more accurate solution is obtained. Thus, the proposed Algorithm 2 achieves the lowest maximum link utilization rate in seven comparison algorithms.

Table 2 shows the average running time of different algorithms. The two routing algorithms we propoesd need longer running times than other traditional algorithms. The MOPSO algorithm needs multiple iterations to find the optimal solution. Moreover, our algorithm adds the path evaluation step after selecting the optimal k paths, which is not mentioned by other algorithms before. The improved MOPSO algorithm in the proposed Algorithm 2 improves the search performance in convergent speed when optimizing the link weights. The proposed Algorithm 2 takes less time than the proposed Algorithm 1. The two algorithms we proposed require more computation and increase the computational complexity of the control plane. However, with the improvement of computing power of hardware devices and the application of distributed multi-controllers in SDN, the computational complexity of control planes will be alleviated.

In order to assess the overall performance of the seven algorithms, the seven paths calculated from the seven algorithms for each request are evaluated by the comprehensive evaluation model. The evaluation model not only can evaluate the overall performance of the target path but can also effectively evaluate the performance of a single network business based on user preferences, which is conducive to realizing a differentiated service network. We use a numerical value from 0 to 1 to express the value of path evaluation and indicates the overall transmission performance of the selected path. The larger the value, the better the overall performance of the path. The data in Table 3 represent the average value of comprehensive evaluation of each path in the corresponding number of requests when the three network applications have the same importance, namely $W^{\prime }=\left[1/3,1/3,1/3\right]$. The data in Table 4 represent the average value of comprehensive evaluation of each path in the corresponding number of requests when the scheme weight $W^{\prime }=\left[0.4,0.4,0.2\right]$. The data in Table 5 represent the average value of comprehensive evaluation of each path in the corresponding number of requests when the scheme weight $W^{\prime }=\left[0.2,0.3,0.5\right]$. It can be seen from these three tables, as the importance of the three network applications specified by the user is different, the two algorithms we proposed have greater comprehensive evaluation values than other algorithms. The comprehensive evaluation values of the paths selected by the proposed Algorithm 1 and the proposed Algorithm 2 are better than those of the other five algorithms; our proposed algorithms select the path with the largest evaluation value by evaluating the model and the other five algorithms do not take it into account. In the process of weight optimization, the proposed Algorithm 2 makes some improvements in optimizing the link weight for global optimal solution and thus shows better performance than the proposed Algorithm 1.

Moreover, we also conduct an experiment when the size of the network topology node is 80 nodes with 602 links and 150 nodes with 2190 links. We fix the number of requests as 400. Figure 7 shows the performance of the cost of paths under different network sizes. From Figure 7 we can see that under different network sizes, the cost of paths for the seven algorithms tends to be the same. As the size of the network grows, the cost of paths increases. Figure 8 shows the load balance of the network under different network sizes. Figure 9 shows the maximum link utilization of the network under different network sizes. Figure 10 shows the comprehensive evaluation value of paths under different network sizes. From these figures, we can see that a comparison of seven algorithms for each performance index under different network sizes exhibits the same trend. The values of load balance and maximum link utilization of the network decrease as the size of the network increases. Thus, we reach the conclusion that our proposed two algorithms achieve better results and Algorithm 2 has the best results under different network sizes.

## 5. Conclusions

In this paper, we propose two routing algorithms for SR in SDN. Our goal is to reduce the consumption of network resources and optimize the load balance of the network as well as select the optimal path in combination with the business preference and actual situation of the network. First, we use the MOPSO algorithm to optimize the link weight and set two objective functions for load balance and network cost. Then, we choose the first *k* shortest paths among all the paths. Finally, the path with the maximum comprehensive evaluation value is selected from the *k* paths. Furthermore, we propose an improved MOPSO algorithm to optimize the link weight. Simulation results show that our proposed algorithms have better performance than other algorithms in terms of the consumption of network resources, load balance, maximum link utilization. Further, our proposed algorithm can choose the best path according to the user’s preference and obtain larger the value of the performance comprehensive evaluation. Our algorithms tend toward theoretical simulation, and further work will perform experiments on real networks and take into account the network throughput and rejection rate for a more comprehensive algorithm.

## Figures and Tables

**Figure 1 sensors-19-00049-f001:**
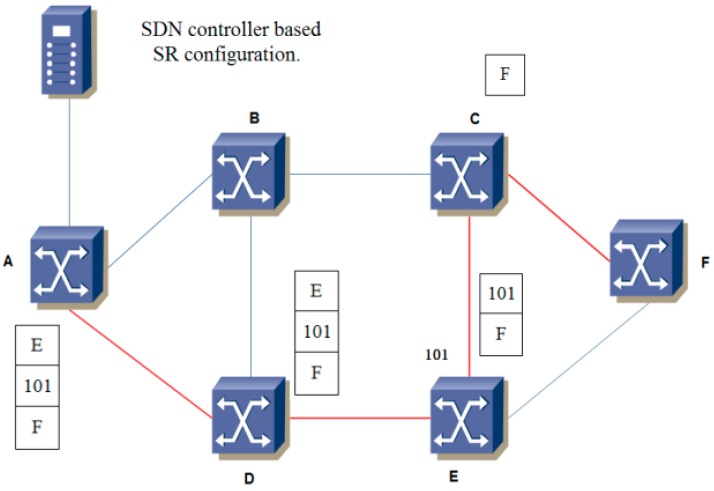
Segment routing based on SDN.

**Figure 2 sensors-19-00049-f002:**
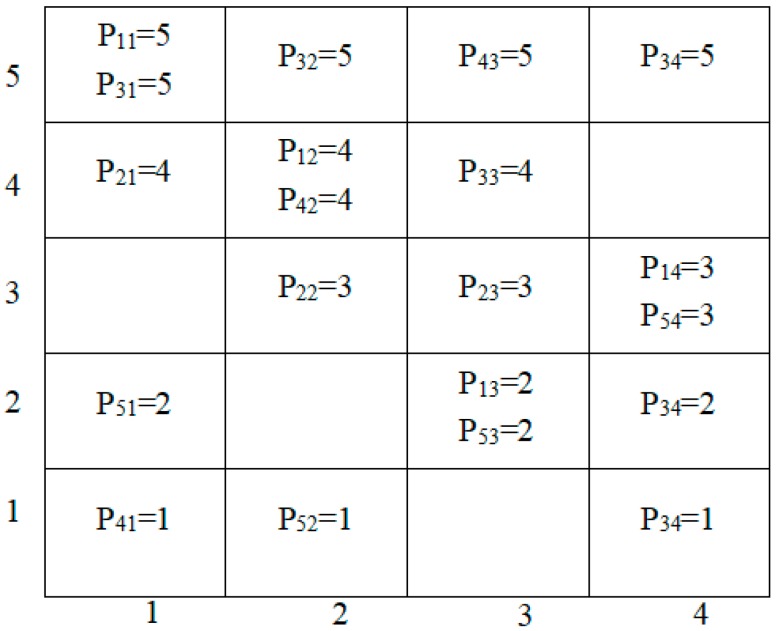
Pareto front in parallel cell coordinate system.

**Figure 3 sensors-19-00049-f003:**
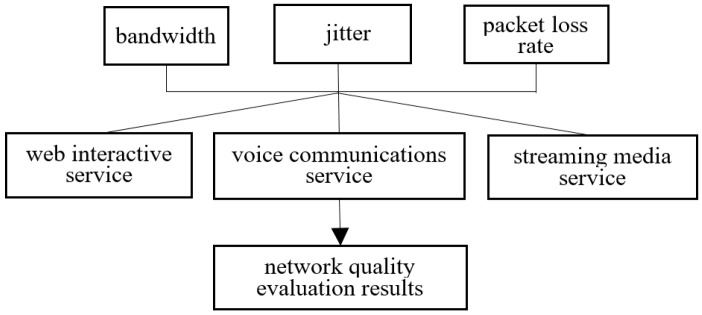
Evaluation hierarchy of Network Performance based on application.

**Figure 4 sensors-19-00049-f004:**
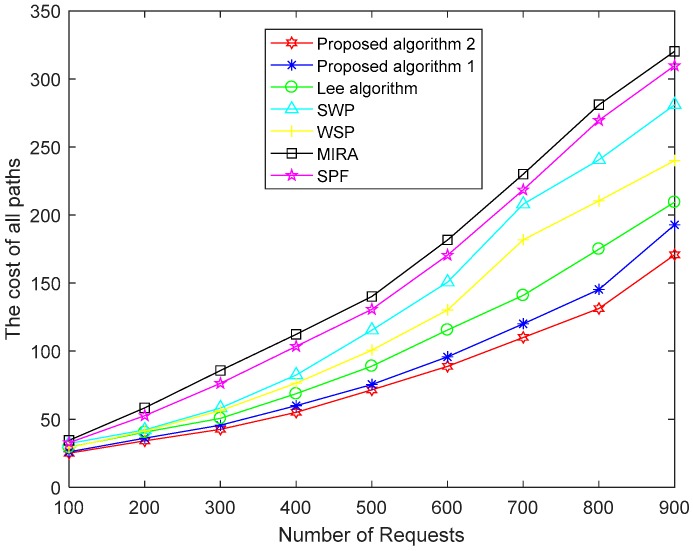
The cost of all paths of the seven algorithms.

**Figure 5 sensors-19-00049-f005:**
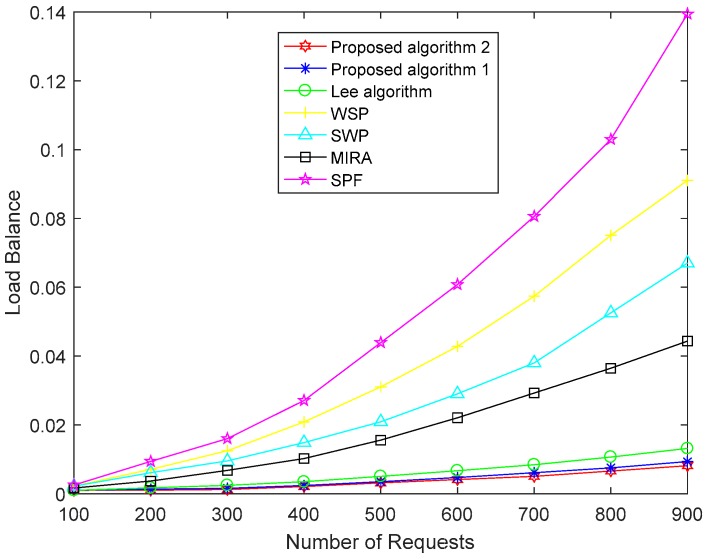
Load balance of the seven algorithms.

**Figure 6 sensors-19-00049-f006:**
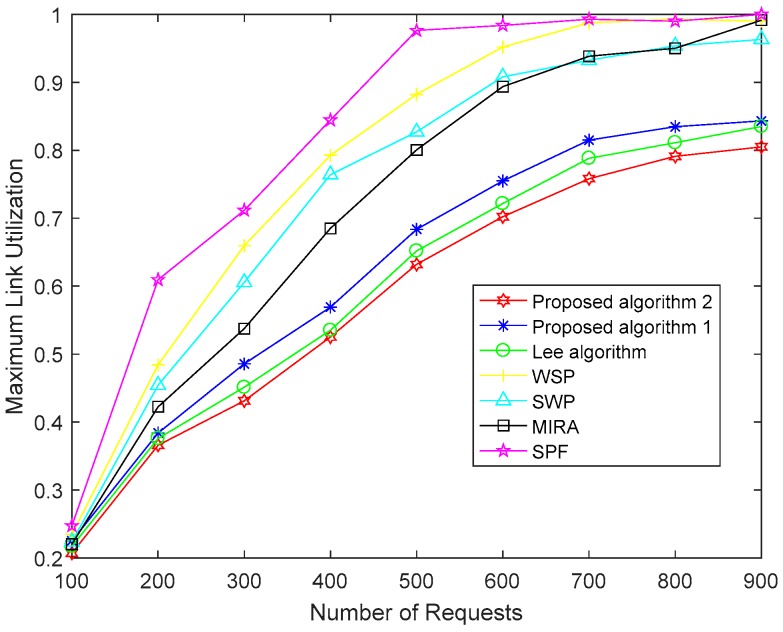
Maximum link utilization of the seven algorithms.

**Figure 7 sensors-19-00049-f007:**
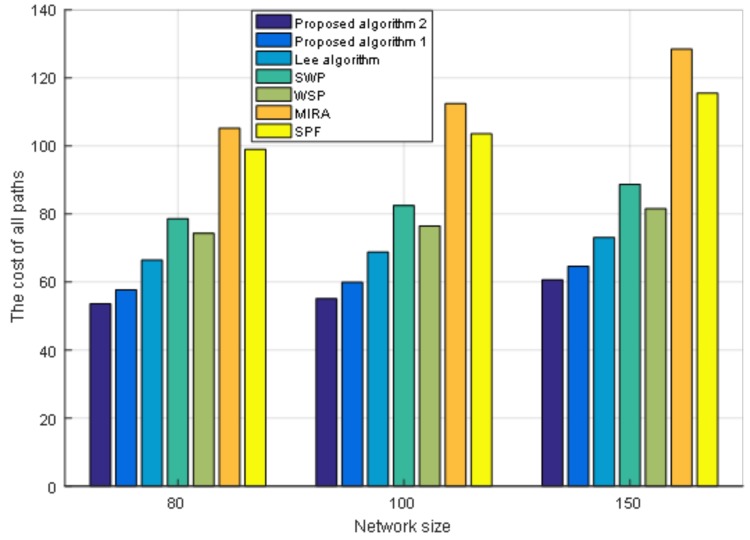
The cost of all paths under different network sizes.

**Figure 8 sensors-19-00049-f008:**
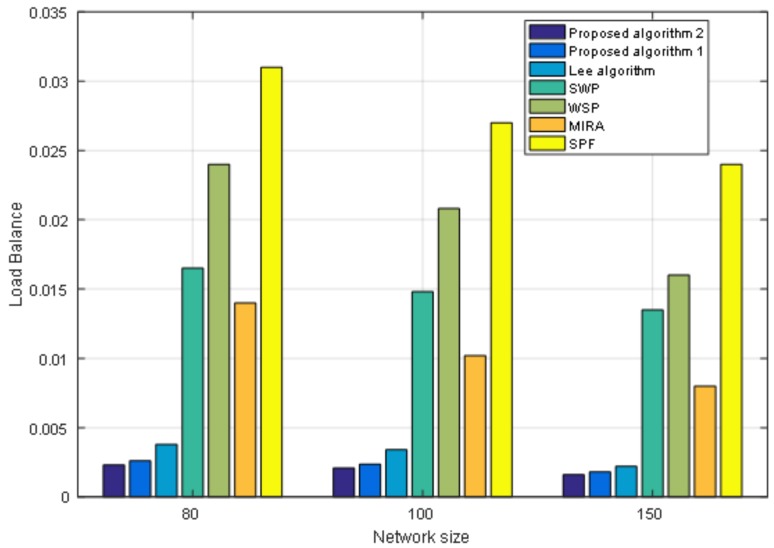
The load balance of network under different network sizes.

**Figure 9 sensors-19-00049-f009:**
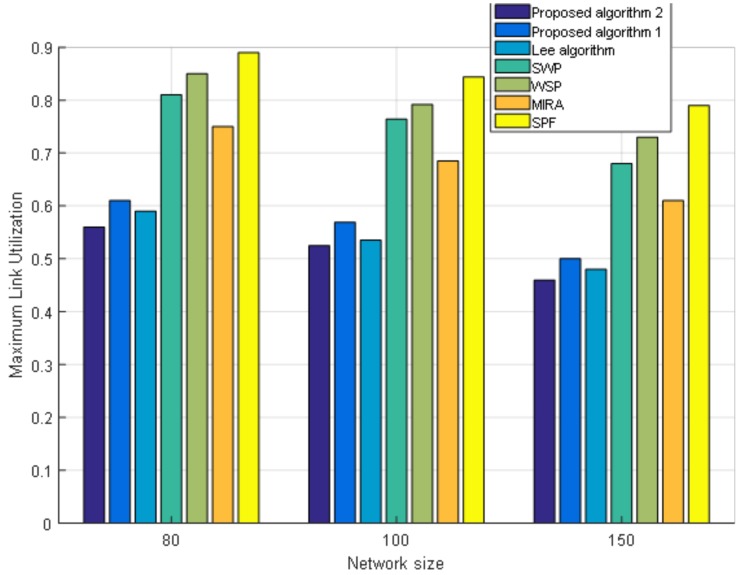
The maximum link utilization under different network sizes.

**Figure 10 sensors-19-00049-f010:**
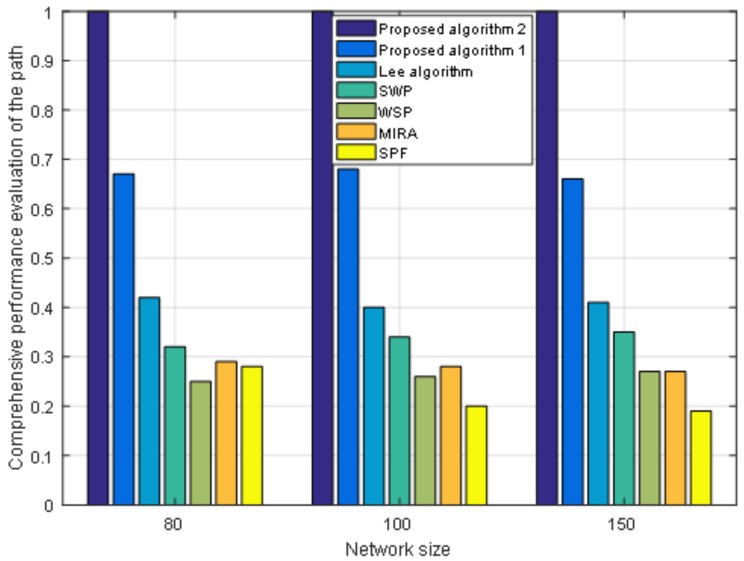
The comprehensive performance evaluation of path under different network sizes.

**Table 1 sensors-19-00049-t001:** Network link measurement data.

Selected Path	Delay/us	Jitter/us	Packet Loss Rate/%
1	6.79	28.87	0.84
2	9.74	45.92	1.81
3	8.96	40.24	1.44
4	8.11	37.28	1.56
5	10.48	49.82	2.22

**Table 2 sensors-19-00049-t002:** The average running time of different algorithms to execute once.

Algorithms	Proposed Algorithm 2	Proposed Algorithm 1	Lee Algorithm	SWP	WSP	MIRA	SPF
running time/s	15.56	10.22	7.16	3.32	3.48	4.12	1.89

**Table 3 sensors-19-00049-t003:** The average value of the comprehensive evaluation path when the scheme weight have the same importance.

Number of Requests	100	300	500	700	900
Proposed Algorithm 2	1.00	1.00	0.99	0.99	0.98
Proposed Algorithm 1	0.63	0.69	0.64	0.70	0.72
Lee algorithm	0.48	0.43	0.33	0.39	0.32
SWP	0.31	0.31	0.35	0.28	0.26
WSP	0.30	0.32	0.22	0.33	0.23
MIRA	0.25	0.21	0.31	0.27	0.29
SPF	0.30	0.25	0.15	0.11	0.15

**Table 4 sensors-19-00049-t004:** The average value of the comprehensive evaluation path when the scheme weight $W^{\prime }=\left[0.4,0.4,0.2\right]$.

Number of Requests	100	300	500	700	900
Proposed Algorithm 2	0.99	1.00	1.00	0.99	0.98
Proposed Algorithm 1	0.68	0.69	0.70	0.68	0.75
Lee algorithm	0.45	0.34	0.38	0.33	0.36
SWP	0.28	0.34	0.30	0.26	0.29
WSP	0.23	0.30	0.28	0.29	0.23
MIRA	0.29	0.26	0.26	0.27	0.25
SPF	0.30	0.22	0.10	0.17	0.11

**Table 5 sensors-19-00049-t005:** The average value of the comprehensive evaluation path when the scheme weight $W^{\prime }=\left[0.2,0.3,0.5\right]$.

Number of Requests	100	300	500	700	900
Proposed Algorithm 2	1.00	0.98	0.99	0.99	1.00
Proposed Algorithm 1	0.73	0.67	0.66	0.70	0.73
Lee algorithm	0.47	0.40	0.37	0.36	0.32
SWP	0.27	0.31	0.32	0.20	0.25
WSP	0.29	0.30	0.24	0.31	0.25
MIRA	0.27	0.25	0.26	0.31	0.33
SPF	0.23	0.26	0.14	0.10	0.17

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
