# Peer review of "An Optimization Routing Algorithm Based on Segment Routing in Software-Defined Networks"

_sensors, 2018, doi:10.3390/s19010049_

Reviewer 1 Report

The paper presents two algorithms to efficiently utilize and load balance traffic across links in a network that is driven via Software-Defined Networking (SDN) and Segment Routing (SR). Authors describe how they exploited SDN and SR in their algorithms and, through extensive simulations, demonstrate how these algorithms minimize path cost (a self-derived metric), achieve better load balancing, and minimize maximum link utilization compared to prior approaches (like Lee's algorithm).

Traffic engineering (like many other problems) is still an unsolved problem in networks, and it's great to see novel approaches to tackle this problem that exploit new trends (e.g., SDN and SR) as they emerge. Though the problem is interesting, I think, the approach---as presented in the paper---lacks in addressing three key issues:

The proposed algorithms only have marginal improvements over the current state-of-the-art algorithm (i.e., Lee's algorithms). The paper fall short in convincing me whether these marginal improvements are that significant in a real setting.

The paper only evaluates improvements in the network (i.e., the data plane). There are no control-plane evaluations, i.e., how much time the algorithms take to compute, etc. When considering a logically-centralized environment (like SDN), control plane quickly becomes a primary bottleneck as the network scales.

SR is designed to mitigate the need for a control plane (as in SDN) and let end-hosts dictate which route packets should take in a source-routed fashion. Therefore, it's confusing when the paper talks about merging SR with SDN. It seems like the approach, presented in the paper, would bring back the limitations that SR is trying to solve. It'd be nice to see if these algorithms can be deployed in a distributed fashion at the end-hosts.

Minor nits:

There is heavy use of passive voice, making it hard to link who is doing what in the sentences.

The term "cost of a path" is confusing. Authors do make it clear later in the paper, but it's not at all clear when reading the abstract and intro.

"actualize differentiated business networks. Not sure what this phrase means.

"and efficient configuration for the upper layer." What does "upper layer" mean?

"needs of new business." What business? Give examples and citations.

"a certain network performance guaranteed." guaranteed --> guarantees.

Section 2.1 is on Segment Routing but also talks about SDN ... confusing. Have a separate section on SDN or update the title accordingly.

"We summarize some of the research on this issue in the following paragraphs. The ..." This sentence is already a part of that paragraph; made me wonder if you are talking about the paragraph after this one; which isn't the case?

"list at the previous first hop router." Is it only for the first-hop router that precedes a switch or any router?

"As SDN has a global view of the network resource, it can monitor network traffic at any time, ..." This is an extreme assertion, without any accompanying references.

"The traffic matrix G1 can be updated by the controller at intervals." What's the frequency of these intervals. Give numbers or citations?

"each flow packet is in the range of 10 MB/s to 100 MB/s." Confusing, how can a packet be in units of MB/s. Did you mean MB? Still odd.

Author Response

key issues:

Point 1: The proposed algorithms only have marginal improvements over the current state-of-the-art algorithm (i.e., Lee's algorithms). The paper fall short in convincing me whether these marginal improvements are that significant in a real setting.

Response 1: Our proposed algorithms have some improvements in network performance (path cost, load balance, maximum link utilization). More importantly, the proposed algorithms have achieved obvious results in path comprehensive evaluation compared with other traditional algorithms. Our algorithms can select the most suitable path according to the needs of different users based on optimizing the basic performance of the network. Firstly, our proposed algorithms select k paths for the purpose of optimizing network performance, and then considers the users need to select one path from k paths. Most of the existing routing algorithms are either considered from the perspective of optimizing network performance(load balance, link utilization,etc) or from the perspective of users’ requirements(packet loss rate, delay, bandwidth,etc). Our algorithms not only optimize the network performance, but also consider the user requirements from the users’ perspective. 

Point 2: The paper only evaluates improvements in the network (i.e., the data plane). There are no control-plane evaluations, i.e., how much time the algorithms take to compute, etc. When considering a logically-centralized environment (like SDN), control plane quickly becomes a primary bottleneck as the network scales.

Response 2: The evaluation of the control plane of the algorithms have been supplemented in Table 4 by taking the running time as an example. It's true that our algorithms add a little bit of computational complexity to the control plane. However, with the improvement of computing power of hardware devices and the application of distributed multi-controllers in SDN(we are studying this aspect), the computational complexity of control planes will be alleviated. We have already supplemented this point in the section 3.3 algorithm summary and in the section 4.2 simulation results. And how to design a multi-targets algorithm with lower complexity and faster speed is also a problem to be solved in the future.

Point 3: SR is designed to mitigate the need for a control plane (as in SDN) and let end-hosts dictate which route packets should take in a source-routed fashion. Therefore, it's confusing when the paper talks about merging SR with SDN. It seems like the approach, presented in the paper, would bring back the limitations that SR is trying to solve. It'd be nice to see if these algorithms can be deployed in a distributed fashion at the end-hosts.

Response 3: Our proposed routing algorithms in the paper are designed for SR in SDN. The SDN controller only needs to code end-to-end routing information into an ordered list of tags and store it in the packet header. Routing is performed by the ordered list of tags in the packet header. And SR avoids the requirement for millions of tag encodings to be stored along each path in each network device and eliminates the complexity of maintaining a mass of forwarding rules. Of course, our method can also be applied to traditional distributed networks. We have already supplemented this point in the section 3.3 algorithm summary.

Minor nits:

Point 1:There is heavy use of passive voice, making it hard to link who is doing what in the sentences.

Response 1: The passive voice in the sentences has been modified. Such as the 33, 57, 118, 126, 128, 146, 178, 185, 236, 258, 275, 295, 296, 297, 309, 353, 354, 394, 553 lines, etc.

Point 2:The term "cost of a path" is confusing. Authors do make it clear later in the paper, but it's not at all clear when reading the abstract and intro.

Response 2: The term "cost of a path" has been supplemented in the 303 to 307 lines of section 3.1.

Point 3:"actualize differentiated business networks. Not sure what this phrase means.

Response 3: "actualize differentiated business networks." have been supplemented by examples in the 41 to 47 lines.

Point 4:"and efficient configuration for the upper layer." What does "upper layer" mean?

Response 4: "upper layer" means the upper application layer. We have modified it in the 29 lines.

Point 5:"needs of new business." What business? Give examples and citations.

Response 5: "needs of new business."have been supplemented by examples in the 41 to 47 lines.

Point 6:"a certain network performance guaranteed." guaranteed --> guarantees.

Response 6: We have changed guaranteed to guarantees.

Point 7:Section 2.1 is on Segment Routing but also talks about SDN ... confusing. Have a separate section on SDN or update the title accordingly.

Response 7: We have updated the title to “Segment Routing based on SDN”.

Point 8:"We summarize some of the research on this issue in the following paragraphs. The ..." This sentence is already a part of that paragraph; made me wonder if you are talking about the paragraph after this one; which isn't the case?

Response 8: We have updated the sentences "We summarize some of the research on this issue in the following paragraphs” to “Some works have been done on this issue”.

Point 9:"list at the previous first hop router." Is it only for the first-hop router that precedes a switch or any router?

Response 9: We have updated the sentences "list at the previous first hop router." to “ list at the previous first hop or segment changed router”.

Point 10:"As SDN has a global view of the network resource, it can monitor network traffic at any time, ..." This is an extreme assertion, without any accompanying references.

Response 10: "it can monitor network traffic at any time." has been deleted.

Point 11:"The traffic matrix G1 can be updated by the controller at intervals." What's the frequency of these intervals. Give numbers or citations?

Response 11: "The traffic matrix G1 can be updated by the controller at intervals." has been updated “The traffic matrix G1 can be updated by the controller after each routing request”

Point 12:"each flow packet is in the range of 10 MB/s to 100 MB/s." Confusing, how can a packet be in units of MB/s. Did you mean MB? Still odd.

Response 12: "each flow packet is in the range of 10 MB/s to 100 MB/s."has been updated "each flow packet is in the range of 10 MB to 100 MB.

Reviewer 2 Report

The paper proposes two metric optimization algorithms to balance the traffic on communication networks based on segment routing. Proposed algorithms are built on MOPSO algorithm, thus the novelty of the paper is not high, howere the author proposed (well-described) effective extension of the reference algorithm. The results clearly shows the benefit of the proposed schemes in terms of path cost, load balance, and link utilization. 

What is not very clear is the connection between the proposed algorithms abd the segment routing technology. At best of my understanding the proposed schemes could be applied also on traditional communcations network not using segment routing. The authors should better clarify this point. 

Author Response

Point 1: What is not very clear is the connection between the proposed algorithms abd the segment routing technology. At best of my understanding the proposed schemes could be applied also on traditional communcations network not using segment routing. The authors should better clarify this point.

Response 1: Our proposed routing algorithms in the paper is designed for SR in SDN. The SDN controller only needs to code end-to-end routing information into an ordered list of tags and store it in the packet header. Routing is performed by the ordered list of tags in the packet header. And SR avoids the requirement for millions of tag encodings to be stored along each path in each network device and eliminates the complexity of maintaining a mass of forwarding rules. Of course, our method can also be applied to the networks not using SR as a general routing algorithm. We have already supplemented this point in the section 3.3 algorithm summary.

Round  2

Reviewer 1 Report

The authors have made a good attempt at addressing the concerns I raised in my earlier comments (esp., Table 4 with runtime results). Though, I am still not convinced about the usefulness of the algorithms proposed in this paper, given that they only provide marginal improvements over existing algorithms with roughly twice the control-plane overhead. Authors state that their algorithms have a better path comprehensive evaluation, but it's not clear how well this evaluation reflects the real environment. 

I believe, authors should add some details on the utility of their proposed algorithms beyond just stating that these algorithms have improved comprehensive evaluation.

Minor nits:

- "and SDN controller 29 provides simple and efficient configuration for the upper application layer [2]" -- what does upper application layer signify here?

-  "In actual networks services ..." -- this is confusing; does this mean that some non-actual services also exist? 

- "Some works have been done on this issue The authors in ..." -- missing dot after issue.

- "The traffic matrix G1 can be updated by the controller after each routing request." -- how frequent are these requests? State numbers and provide citations.

- "single evaluation indexand achieve ..." -- missing space after index.

- "Of course, our method can also be applied to traditional distributed networks ..." -- provide some simple directions to guide the reader on how this would be done.

- In Table 4 "running time/s" -- what does this term mean? It seems like a unitless number as time is being divided by time. Maybe you meant, "running time (s)"?

Author Response

I believe, authors should add some details on the utility of their proposed algorithms beyond just stating that these algorithms have improved comprehensive evaluation.

 Response:We have added a more detailed description in lines 413 to 424 in Section 3.3.

Minor nits::

Point 1: "and SDN controller 29 provides simple and efficient configuration for the upper application layer [2]" -- what does upper application layer signify here?

Response 1:

As a new network architecture, Software Defined Network (SDN) redefines and abstracts forwarding planes, device control planes, and application layer, making network device software programmable. Bottom-up, the SDN architecture is divided into three layers: infrastructure layer, control layer and application layer. The control layer abstracts the resources of the underlying network infrastructure, provides a global network abstract view for the upper layer application, and is implemented by software to get rid of the bundle of network control functions by the hardware network device. The application layer programs the network abstraction provided by the control layer through the open interface provided by the control layer to control the network traffic of various traffic models and applications, so that the traffic generated by the application is aware of the network and realizes network intelligence.

“The upper application layer” we mentioned means the application layer in SDN architecture. SDN controller is responsible for the control layer in the SDN network.

Point 2:  "In actual networks services ..." -- this is confusing; does this mean that some non-actual services also exist?

Response 2: “actual” is an expression error and has been deleted. We want to express the general network services, no other extra meaning.  

Point 3: "Some works have been done on this issue The authors in ..." -- missing dot after issue.

Response 3:  The dot has been added.

Point 4: "The traffic matrix G1 can be updated by the controller after each routing request." -- how frequent are these requests? State numbers and provide citations.

Response 4:  Routing requests are sent 10 times per second and has been marked in line 453 in section 4. The citations has been added.

Point 5:  "single evaluation indexand achieve ..." -- missing space after index.

Response 5:  The missing spaces have been added.

Point 6:  "Of course, our method can also be applied to traditional distributed networks ..." -- provide some simple directions to guide the reader on how this would be done.

Response 6: We have added these in lines 450 to 452 in section 3.3: In the traditional distributed networks, our proposed routing algorithm can be applied in the routing calculation module of router by online or offline.

Point 7:  In Table 4 "running time/s" -- what does this term mean? It seems like a unitless number as time is being divided by time. Maybe you meant, "running time (s)"?

Response 7: We have changed "running time/s" to "running time/sec". The “running time” is the average time that the algorithms is executed once, and we have modified it in the title.

“sec” represents that the unit of the number in the table is seconds.